

# A Modular, Non-Newtonian, Model, Library Framework (DebrisLib) for Post-Wildfire Flood Risk Management

Ian E. Floyd[1], Alejandro Sanchez[2], Stanford Gibson[2], and Gaurav Savant[3]

[1]Research Physical Scientist, U.S. Army Engineer Research and Development Centre, Vicksburg, 39180, U.S
[2]Senior Research Engineer, U.S. Army Corps of Engineers Hydrologic Engineering Centre, Davis, 95616, U.S.
[3]Research Hydraulic Engineer, U.S. Army Engineer Research and Development Centre, Vicksburg, 39180, U.S

*Correspondence to*: Ian E. Floyd (Ian.E.Floyd@usace.army.mil)

**Abstract.** Wildfires increase flow and sediment load through removal of vegetation, alteration of soils, decreasing infiltration, and

production of ash commonly generating a wide variety of geophysical flows (i.e., hyperconcentrated flows, mudflows, debris

flows, etc.). Numerical modellers have developed a variety of Non-Newtonian algorithms to simulate each of these processes, and

therefore, it can be difficult to understand the assumptions and limitations in any given model or replicate work. This diversity in

the processes and approach to non-Newtonian simulations makes a modular computation library approach advantageous. A

computational library consolidates the algorithms for each process and discriminates between these processes and algorithms with

quantitative non-dimensional thresholds. This work presents a flexible numerical library framework (DebrisLib) to simulate large-

scale, post-wildfire, non-Newtonian geophysical flows using both kinematic wave and shallow-water models. DebrisLib is derived

from a variety of non-Newtonian closure approaches that predict a range of non-Newtonian flow conditions. It is a modular code

designed to operate with any Newtonian, shallow-water parent code architecture. This paper presents the non-Newtonian model

framework and demonstrates its effectiveness by calling it from two very different modelling frameworks developed by the U.S.

Army Corp of Engineers (USACE), specifically, within the one-dimensional and two-dimensional Hydrologic Engineering Centre

River Analysis System (HEC-RAS) and two-dimensional Adaptive Hydraulics (AdH) numerical models. The development and

linkage-architecture were verified and validated using two non-Newtonian flume experiments selected to represent a range of non-

Newtonian flow conditions (i.e., hyperconcentrated flow, mudflow, debris flow) commonly associated with post-wildfire flooding.

## 1 Introduction

The number and intensity of large wildfires is a growing concern in the United States. Over the past decade, the National

Interagency Fire Centre (NSTC, 2015) reported more large fires in every western state in the arid and semi-arid western region of

the country, than any previous decade. Wildfires remove vegetation, reduce organic soil horizons to ash, extirpate microbial

communities, and develop hydrophobic soils. These processes all increase water and sediment runoff. Post-wildfire environments

can cause a spectrum of hydrologic and sedimentation responses ranging from minor runoff events to catastrophic floods and

deadly debris flows. The high sediment concentration exacerbates damages from these events, which have been documented



around the world (Rowe et al., 1954; Lane et al., 2006; Shin, 2010; Shakesby, 2011; Moody et al., 2013). In the years following a wildfire, ecotone shifts, gully formation, and channel incision alter the hydrologic system response, resulting in dramatic, and sometimes prolonged changes to downstream systems. Wildfires impact hydrology by removing the rainfall interception canopy, generating ash and charred material, reducing organic binding materials in soils, increasing hydrophobic soils, and modifying the

intrinsic and hydraulic properties of soils and sediments (Certini, 2005; Moody et al., 2009; Ebel et al., 2012). Lahars and mine tailing dams also threaten communities and infrastructure around the world and, like post-wildfire flows, these events diverge from classic, Newtonian, shallow-water flow assumptions.

Researchers have developed a range of numerical models to simulate non-Newtonian flows for hazard assessment, flood risk evaluation, and mitigation design (O'Brien et al., 1993; Iverson and Denlinger, 2001; Hungr et al., 2005). These physics-based

numerical models simulate advection with constitutive laws of fluid mechanics in one- and two-dimensions. Savage and Hutter (1989) provided the foundation for predicting non-Newtonian flow using Saint-Venant based shallow-water equations which have been commonly applied for numerical modelling of rapid mass movements over irregular geometry (e.g., Chen and Lee, 2000; Iverson and Denlinger, 2001; Imran et al., 2001; Pudasaini et al., 2005; Naef et al., 2006; Martinez et al., 2007; Pastor et al., 2009; Luna et al., 2012).

There is an increasing need to develop a post-wildfire modelling framework and enhance non-Newtonian numerical modelling capabilities within hydrologic and hydraulic engineering models, for decision making in emergency management and flood risk management operations. However, with the diversity of post-wildfire processes (floods, mudflows, debris flows, etc.) and multiple algorithms available to simulate each process, a formal, modular, library framework that consolidates these algorithms and provides them to multiple hydrodynamic codes will make non-Newtonian implementation more transparent, repeatable, and robust as

demonstrated here. The non-Newtonian algorithm library (DebrisLib) provides a flexible, modular, non-Newtonian, numerical-modelling framework that developers can call from any one-dimensional or two-dimensional shallow-water-based hydraulic and hydrologic model. Consolidating multiple non-Newtonian closures associated with the continuum of geophysical flow processes in a modular library and sharing it between hydrodynamic codes has three main advantages:

1.  The library consolidates diverse literature, making a wide suite of algorithms available to each linked model, avoiding

55       duplication of effort.

2.  The library makes non-Newtonian modelling more transparent by standardizing algorithm implementation.

3.  The library leverages the validation and verification activities of multiple development communities, accelerating the quality control of the code.

This paper documents the numerical library development, enhancements, and linkage architecture necessary for predicting post-
wildfire non-Newtonian flood events across different numerical modelling libraries, specifically one- and two-dimensional USACE
numerical models. Subsequent papers will look at application of this work to real-world post-wildfire conditions.

## 2 Background

Wildfires can significantly shock or perturb semi-arid and arid environments (Pierce et al., 2004; Orem and Pelletier, 2016). Most
western U.S. watersheds require decades to recover. These events increase flow and sediment loads, creating immediate and long-
term management concerns for federal, state, and local agencies. These management challenges include predicting the post-wildfire
flood response of a particular wildfire-precipitation mix. System responses can range from minimal impact, to increased clear
water flows, to more destructive ash flows, mud floods, and debris flows. Mudflows, debris flows, and ash flows are unsteady
gravity-driven events that involve complex mixtures of sediment, water, and entrained material (i.e., organics, woody debris,
unconsolidated substrate). These non-Newtonian flows have high sediment concentrations and can carry large boulders, trees, and
uprooted infrastructure (e.g., upstream structures or bridge debris). These flows are commonly modelled using shallow-water
equations as either 1) single phase, 2) two-phase, or 3) mixture theory, using non-Newtonian closure approaches and
approximations (e.g., Iverson, 1997; Jin and Fread, 1997; Imran et al., 2001). The grain-size distribution, sediment concentration,
and flow stress state (as a function of slope) determine which geophysical flow particular wildfire-precipitation events generate.

In addition to the process complexity, modellers face algorithmic diversity challenges. Each of these processes have multiple
modeling approaches. Researchers have developed a wide range of algorithms to help apply these rheological and geotechnical
modeling libraries to different classifications of geophysical flows. The methods that simulate these post-wildfire flood events are
commonly grouped into four main categories:

1) Linear (Bingham 1922) and non-linear viscoplastic models (Jin and Fread 1997; Imran et al. 2001). These approaches use
rheological modes as heuristics for cohesive and/or viscous stresses in the fluid and are commonly used to describe the rheology
of laminar mud flows.

2) Dispersive-turbulent stress models (O'Brien et al. 1993). This approach adds a turbulent stress term to the dispersive model to
describe the inter-particle mechanics within a clay, silt, and organic matrix. This model is commonly applied to sediment mixtures
containing cohesive sediment in quantities greater than 10% by volume.

3) Dispersive fluid models (Bagnold 1954 and 1956; Takahashi 1978). The dispersive models use Bagnold's theory to describe the
non-linear fluid stress that develops from particle-to-particle interaction between granular, clastic (i.e., silt, sand, or gravel)
particles, where the fluid fluctuations keep the particles suspended. This model is commonly applied to sediment-water mixtures
containing mostly sand and gravel with lower quantities of fine sediment ($\leq 5\%$–10% by volume).





4) Coulomb-based frictional models (Hungr 1995; Iverson 1997; Iverson and Denlinger 2001). Friction dominated models like Coulomb or Vollemy (Iverson, 1997) take a more geotechnical approach to simulate grain flows that approach library supported

conditions, where the internal mixture stresses are dominated by inter-granular friction. These approaches are often applied to heterogeneous, poorly sorted (well graded), clastic debris flows approaching land-slide classifications.

Non-Newtonian flows include several regimes (or flow states) depending on the solid concentration in the fluid and the grain size of the sediment. In general, as concentration increases and grain size coarsens, the fluid-sediment mixture passes through five classifications: 1) Newtonian Flow (clear water or alluvial sediment transport), 2) Hyperconcentrated flow, 3) Mudflow, 4) Grain

Flow (coarse-grain hyperconcentrated flow), and 5) Debris (clastic) flow. A taxonomy and conceptual model were developed to simplify the non-Newtonian flow continuum onto a single axis as a function of concentration and grain size distribution shown below in Figure 1.

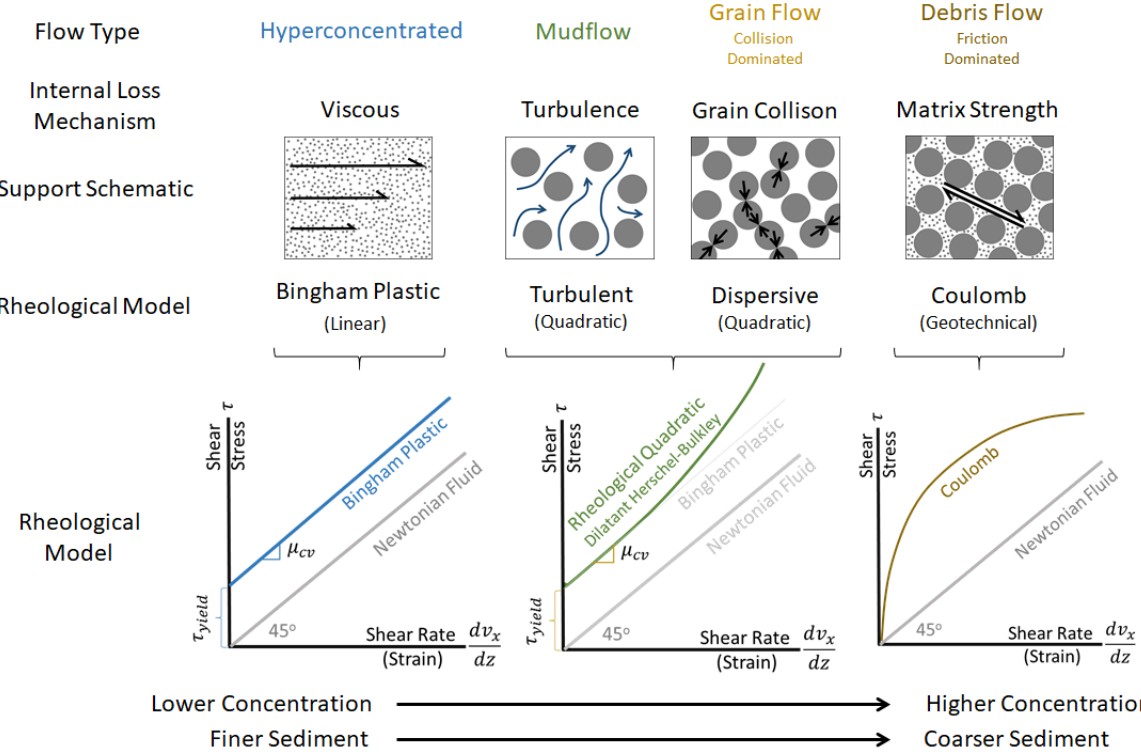

**Figure 1. Classification, processes, conceptual model, and rheological model of the four flow conditions in the non-Newtonian model**
**library.**

The research team developed a non-Newtonian library (DebrisLib) to simulate the continuum of post-wildfire flood responses along these concentration and grain size gradients, based on rheology and Coulomb theory, to operate independently of the shallow-water parent codes. The non-Newtonian library assembles closure approaches that different shallow-water codes can leverage and

determines the flow conditions using non-dimensional threshold parameters. In addition, the library estimates a non-Newtonian drag coefficient (Happel and Brenner, 1965; Julien 2010), addresses hindered settling (Richardson and Zaki, 1954; Baldock et al., 2004; Cuthbertson et al., 2008), computes relative buoyancy terms, and accounts for increased viscosities and mass density (Einstein and Chien, 1955; Dabak and Yucel, 1986). The diversity of the processes and sub-diversity of algorithms for each process makes a shared library approach particularly useful for non-Newtonian model development. Future versions of DebrisLib will address both sediment erosion, transport, and deposition.


Applying non-Newtonian transport in a shallow-water model requires computes internal losses by adding a slope term to the friction slope ($S_f$) in the momentum equation and bulking (or increasing) the flow to account for the volume of the solids. The shallow-water friction slope ($S_f$) accounts for the forces acting against the flow at the fluid boundary (e.g., the channel) while the "mud and debris slope" ($S_{MD}$) accounts for the internal losses due to the viscosity, turbulence, and/or dispersion within the fluid.

The library computes a non-Newtonian shear stress based on the flow classification (e.g., mudflow, debris flow, etc.) and the appropriate rheological approach (i.e., stress-strain model). Then the library integrates the viscous, turbulent, and dispersion shear from the stress-strain model into the momentum equation by converting the shear to a slope and adding this slope to the friction slope. Additionally, because these flows can include between 5% and 70% solids by volume, a fixed bed model must augment the flow volume to account for the impact of the sediment on the mass and depth of the flow.

**3. Methods**

**3.1 Shallow-Water Numerical Models**

The shallow-water flow equations solve the continuity and momentum equations simultaneously to compute water stage and velocity. The frictional forces between the fluid and the solid boundary are the primary resisting forces in the standard Newtonian clear-water hydraulic equations. In one dimension, the conservation of mass and momentum are the Saint-Venant equations:

$$\frac{\partial Q}{\partial x} + \frac{\partial A}{\partial t} - q = 0,  \tag{1}$$

$$\frac{\partial Q}{\partial x} + \frac{\partial Q\bar{u}}{\partial x} + gA\left(\frac{\partial \eta}{\partial x} + S_f + S_{MD}\right) = 0,  \tag{2}$$

where

Q = volumetric flow discharge (m³/s)

x = downstream distance in channel (m)

t = time (s)

A = cross sectional area of channel (m²)

q = lateral inflow or outflow (m²/s)





$\eta$ = water surface elevation (m)

$S_f$ = Newtonian friction slope (m/m).

$S_{MD}$ = Mud and debris friction slope (m/m).

In two dimensions the shallow-water equations vertically integrate the mass and momentum equations under the assumptions of incompressible flow and hydrostatic pressure. Assuming negligible free surface shear and pressure variations at the free surface, the two-dimensional shallow-water equations are:

$$\frac{\partial Q}{\partial t} + \frac{\partial F_x}{\partial x} + \frac{\partial F_y}{\partial y} + H = 0, \tag{3}$$

$$Q = \begin{Bmatrix} h \\ uh \\ vh \end{Bmatrix}, \tag{4}$$

$$F_x = \begin{Bmatrix} uh \\ u^2h + \frac{1}{2}gh^2 - h\frac{\sigma_{xx}}{\rho_m} \\ uvh - h\frac{\sigma_{yx}}{\rho_m} \end{Bmatrix}, \tag{5}$$

$$F_y = \begin{Bmatrix} vh \\ uvh - h\frac{\sigma_{xy}}{\rho} \\ vh + \frac{1}{2}gh^2 - h\frac{\sigma_{yy}}{\rho} \end{Bmatrix}, \tag{6}$$

$$H = \begin{Bmatrix} 0 \\ gh\frac{\partial z_b}{\partial x} + gh\frac{n^2 u\sqrt{u^2+v^2}}{h^{\frac{4}{3}}} + ghS_{MDx} \\ gh\frac{\partial z_b}{\partial y} + gh\frac{n^2 v\sqrt{u^2+v^2}}{h^{\frac{4}{3}}} + ghS_{MDy} \end{Bmatrix}, \tag{7}$$

where

$\rho_m$ = mixture density (kg/m³)

     $u$ = flow velocity in the x direction (m/s)

     $v$ = flow velocity in y direction (m/s)

     $h$ = flow depth (m)

     $\sigma_{ij}$ = Reynolds stresses due to turbulence (Pa)

$S_{MDx}$ = Mud and debris slope in the x direction (m/m)

     $S_{MDy}$ = Mud and debris slope in the x direction (m/m)

Where the first subscript indicates the direction and the second indicates the face on which the stress acts.

     $g$ = gravitational acceleration (m/s²)

     $z_b$ = bottom elevation (m)

$n$ = Manning's roughness coefficient





The Reynolds stresses are determined using the Boussinesq approach to the gradient in the mean current:

$$\sigma_{xx} = 2\rho_m \nu \frac{\partial u}{\partial x}, \tag{8}$$

$$\sigma_{yy} = 2\rho_m \nu \frac{\partial v}{\partial y}, \tag{9}$$

$$\sigma_{xy} = \sigma_{yx} = \rho_m \nu \left(\frac{\partial u}{\partial y} + \frac{\partial v}{\partial x}\right), \tag{10}$$

The depth-averaged Shallow Water Equations model in HEC-RAS solves volume and momentum conservation equations and includes temporal and spatial accelerations as well as horizontal mixing while the diffusive wave equation model ignores these processes but is therefore simpler and more computationally efficient. The 2D volume conservation of the water-solid mixture is given by:

$$\frac{\partial \eta}{\partial t} + \frac{\partial (hu)}{\partial x} + \frac{\partial (hv)}{\partial y} = q, \tag{11}$$

where $\eta$ is the flow surface elevation, $t$ is time, $h$ is the water depth, $V$ is the velocity vector, and $q$ is a source or sink term, to account for external and internal fluxes. The depth-averaged momentum conservation equations may be written as (Hergarten and Robl, 2015):

$$\frac{\partial u}{\partial t} + u\frac{\partial u}{\partial x} + v\frac{\partial u}{\partial y} = -g\cos^2\varphi\frac{\partial \eta}{\partial x} + \frac{1}{h}\frac{\partial}{\partial x}\left(v_t h \frac{\partial u}{\partial x}\right) + \frac{1}{h}\frac{\partial}{\partial y}\left(v_t h \frac{\partial u}{\partial y}\right) - \frac{\tau_x}{\rho_m R}\frac{\cos\psi}{\cos\varphi}, \tag{11}$$

$$\frac{\partial v}{\partial t} + u\frac{\partial v}{\partial x} + v\frac{\partial v}{\partial y} = -g\cos^2\varphi\frac{\partial \eta}{\partial y} + \frac{1}{h}\frac{\partial}{\partial x}\left(v_t h \frac{\partial v}{\partial x}\right) + \frac{1}{h}\frac{\partial}{\partial y}\left(v_t h \frac{\partial v}{\partial y}\right) - \frac{\tau_y}{\rho_m R}\frac{\cos\psi}{\cos\varphi}, \tag{11}$$

in which $g$ is the gravitational acceleration, $v_t$ is a turbulent eddy viscosity, $\tau = (\tau_x, \tau_y)$ is the total basal stress, $\rho_m$ is the water-solid mixture density, $R$ is the hydraulic radius, , $\varphi$ is the water surface slope, and $\psi$ is the inclination angle of the current velocity direction. In the above equations, the second term on the right-hand-side represents the horizontal mixing due to turbulence and also in the case of a debris flow, horizontal mixing due to particle collisions. Utilizing the conservative form of the mixing terms is essential for accurate momentum conservation. The bottom friction coefficient is computed utilizing the

Manning's roughness coefficient as $\tau = \tau_{turbulent} + \tau_{MD}$, where $\tau_{turbulent}$ is the turbulent stress and $\tau_{MD}$ is the mud and debris stress which includes all non-Newtonian stresses. The turbulence bottom shear stress is computed as a function of the Manning's roughness coefficient. The mud and debris stress are described in detail in the section "Rheological Models". It is noted that when the non-Newtonian stress is equal to zero and the cosine functions (slope corrections) are removed, the above 2D shallow-water equations reduce to the clear-water equations utilized in HEC-RAS.

**3.2 Non-Newtonian Shallow-Water Closure**

Mud and debris flows generate additional resisting forces. Increasing the solid content increases the viscosity of non-Newtonian flows, generating internal resisting forces within the fluid. At higher concentrations, particularly with coarse particles, particle





collision and friction introduce additional internal resisting forces. Most of the theoretical and numerical modifications involve integrating the new internal fluid forces in the momentum equation. The depth-averaged equations can be adapted for non-Newtonian simulations by adding an additional loss slope term ($S_{MD}$) to the classic friction slope term ($S_f$) in the conservation of momentum equation.

DebrisLib computes this Mud and Debris slope that the hydrodynamic models add to the momentum equation by computing internal shear stresses that the different non-Newtonian processes generate based on rheology or Coulomb models (see Figure 1). The library then converts the internal, rheological shear ($\tau_{MD}$) from the hypothesized stress-strain characteristics into a fluid loss Mud and Debris slope ($S_{MD}$):

$$S_{MD} = \frac{\tau_{MD}}{\rho_m g R}, \tag{11}$$

where

$\rho_m$ = the sediment-fluid mixture density (kg/m³)

$R$ = the hydraulic radius (m)

The O'Brien et al. (1993) quadratic, rheological model combines the four stress components of non-Newtonian sediment mixtures: (1) cohesion between particles, (2) internal friction between fluid and sediment particles, (3) turbulence, and (4) inertial impact between particles. The quadratic model separates the stress-strain relationships into these four, additive components, such that the shear stress is:

$$\tau_{MD} = \tau_{yield} + \tau_{viscous} + \tau_{turbulent} + \tau_{dispersive}, \tag{12}$$

Where

$\tau_{MD}$ = the total mud-and-debris shear stress (Pa)

$\tau_{yield}$ = yield stress (Pa)

$\tau_{viscous}$ = viscous shear stress (Pa)

$\tau_{turbulent}$ = turbulent shear stress (Pa) (similar to Manning's roughness in O'Brien et al., 1993)

$\tau_{dispersive}$ = dispersive shear stress (Pa)

O'Brien et al. (1993) define these terms, yielding a quadratic model based on the strain ($dv_x/dz$):

$$\tau_{MD} = \tau_y + \mu_m \left(\frac{dv_x}{dz}\right) + \rho_m l_m^2 \left(\frac{dv_x}{dz}\right)^2 + c_{Bd} \rho_s \left(\left(\frac{C_*}{C_v}\right)^{1/3} - 1\right)^{-2} d_s^2 \left(\frac{dv_x}{dz}\right)^2, \tag{13}$$

where,

$dv_x/dz$ = the shear rate (1/s) computed as a function of depth averaged velocity and flow depth





$\mu_m$ = mixture dynamic viscosity (Pa s)

$\rho_m$ = sediment mixture mass density (kg/m³)

$l_m$ = mixing length (m)

$c_{Bd}$ = Bagnold impact empirical coefficient ($c_{Bd} \cong 0.01$)

$\rho_s$ = sediment particle density (kg/m³)

$C_*$= maximum volumetric sediment concentration (-)

$C_v$ = volumetric sediment concentration (-)

$d_s$ = sediment grain size (m)

Takahashi (1980) identified experimentally that the Bagnold impact coefficient ($c_{Bd}$) ranges between 0.35 and 0.5 and that is

significantly larger than the value recommended from Bagnold (1954 and 1956). Iverson (1997) defined the strain (or shear rate)

($3\bar{u}/h$) is based on a vertical integration of a parabolic velocity profile or ($2\bar{u}/h$) for linear velocity profile conditions, where,

$\bar{u}$ = depth averaged velocity (m/s)

$h$ = flow depth (m).

Therefore, the quadratic model requires two new terms: the mixture density and the Prandtl mixing length. The equation for the

Prandtl mixing length is defined as

$$l_m = kz, \tag{14}$$

where

$k$ = the Von Karmen coefficient ($\cong 0.41$)

$z$ = the proportional distance from the boundary (bed).

This quadratic model combines linear and non-linear rheological modes to compute internal shear. Rheological models do not

perform as well as the mixtures get more clastic (i.e., high concentrations of coarse particles). DebrisLib simulates clastic debris

flows with a Coulomb approximation based on the Johnson and Rodine (1984) Coulomb viscous model (Naef et al., 2006). This

approach replaces the Bingham yield strength ($\tau_y$) a geotechnical, Coulomb yield stress defined as,

$$\tau_{yc} = \rho_m g h \cos\alpha \tan\varphi, \tag{15}$$

where,

$\tau_{yc}$ = Coulomb yield stress (Pa)

$\alpha$ = bed slope angle (°)

$\varphi$ = Coulomb friction angle (°) with typically ranges between 30° and 40° (Iverson, 1997; McArdell et al., 2007).



The Coulomb friction angle is a function of the individual grain friction angle and the packing geometry of the particles along the

failure plane.

### 3.3  Numerical Model Discretization

The solution of Equations 1 thru 15 can be achieved through discretization in time and space. Temporal discretization of the

equations is achieved through explicit or implicit solution schemes. Explicit schemes rely on the solutions from the previous time

steps to obtain the new time step solution, and implicit schemes rely on the previous, present and future solutions to obtain the new

time step solution. Numerical techniques for the spatial discretization include finite difference method (FDM), Finite Volume

method (FVM), Finite Element method (FEM) and mesh less methods. In this manuscript we used the FEM and the FVM for the

spatial discretization of the afore-mentioned equations. We used the USACE 1D and 2D HEC-RAS and 2D AdH models to test

the robustness of the developed techniques. The HEC-RAS solves the diffusive wave equation (DWE) and the depth-averaged

shallow water equations (SWE). The DWE is solves using an implicit FVM. The SWE are solved with a combination of finite

difference and finite volume methods and a semi-implicit time stepping scheme. Water volume conservation is ensured by a finite

volume discretization of the continuity equation. The momentum equation is discretised with semi-implicitly using the FDM.

HEC-RAS uses a subgrid modelling approach which describes the high-resolution subgrid terrain using hydraulic property tables

allowing for larger computational cells and time steps while still maintaining accuracy. The subgrid approach leads to a mildly

nonlinear system of equations which is solved using a Newton-type iteration algorithm (HEC 2016). AdH suite is a collection of

solvers for the unsaturated Richard's equations, Reynolds Averaged Navier Stokes Equations (RANS), full momentum shallow

water equations (FMSWE) and the DWE. The FMSWE and the DWE can be meshed in space using the Cartesian or the spherical

coordinate system. AdH uses the FEM method with implicit time stepping to solve the equations of motion and conservation of

mass. AdH has been widely discussed in engineering literature and the interested reader is referred to Savant et al. (2019), Savant

et al. (2018), Trahan et al. (2018), Savant and McAlpin (2014), Savant and Berger (2012), and Savant et al. (2011) for additional

details on the AdH model. HEC-RAS solves the FMSWE and the DWE using a combination of implicit finite difference and finite

volume methods on an unstructured polygonal mesh.

### 3.4  Verification and Validation Datasets

The verification and validation process involved simulation of multiple flume experiment selected to represent the continuum of

non-Newtonian flow behaviour under both steady and unsteady conditions. This includes the Jeyepalan (1981) and Hungr (1995)

unsteady dam break analytical solutions, Haldenwang (2003) steady state flume experiments, and the large-scale US Geological

Survey (USGS) flume experiments of high concentration debris flow conditions from Iverson et al. (1992 and 2010). Initial

verification of DebrisLib was conducted using the 1D analytical solutions from an idealized 30-meter-high tailings dam subjected





to instant liquefaction failure from Jeyapalan (1981) and Hungr (1995). The primary verification variables included failure runout

distance, velocity, depth, and cessation of flow. The initial validation flume experiments are from Haldenwang et al. (2006) who

conducted flume experiments using kaolinite, bentonite, and solution of carboxymethyl cellulose polymer with various

concentration ranging from 1% to 10% by volume. The experiments were conducted in two tilting flumes, one 10-m long and 300-

mm wide and the other flume 5-m long and 75-mm wide. Slope varied ranging from 1º to 5º. In this paper only the results from

the 10-m flume are presented. The final experiments utilized were the United State Geological Survey (USGS) debris flow flume

at H.J. Andrews Experimental Forest, Oregon, United States (Iverson et al., 1992; Iverson 1992; Iverson et al., 2010). The USGS

conducted large-scale debris flow experiments between 1994 and 2004. The USGS flume experiments consisted of rapid releases

of saturated nonuniformly sized sediment mixtures (Iverson et al. 2010). Experiments where conducted in a 95-m-long, 2-m-wide

flume with a maximum slope of approximate 31º for both fixed- and mobile-bed conditions for a wide range of sediment gradations

and concentrations. The headgate is positioned at 12.5 m downslope. This section leads to a reach in which the slope follows a

catenary curve descending 2.2 m vertically before reaching an outlet to a runout surface with a gentle slope of 2.4º extending

107.5 m. The rough concrete tiles are positioned between 6 to 79 m from the release gate. The height of debris behind the headgate

was 1.9 m. Simulations presented here were conducted for fixed-bed experiments using debris mixtures of about 56% gravel, 37%

sand, and 7% mud particles (Iverson et al., 2010). The HEC-RAS grid for the USGS flume experiment has a constant grid resolution

of 0.15 and 0.2 in the downslope and transverse directions respectively with a total of approximately 13,350 cells.

**4 Results**

The model library and the linkage-architecture were evaluated by simulating two non-Newtonian verification data sets by calling

the library from two different hydrodynamic codes. To evaluate the model library a selection of flume experiments was used to

demonstrate the library's flexibility using both small- and large-scale flows of various grain size distributions and sediment

concentrations. These flume experiments were selected to demonstrate the flexibility of the model library and its linkage

architecture with USACE shallow-water models. The data sets simulated include: equilibrium, hyperconcentrated flow (Cv=10%)

from Haldenwang et al. (2006) experiments and an analytical, unsteady non-Newtonian dam break with a higher concentration

(Cv=55%), from Jeyapalan 1981 and Hungr (1995) running out until flow stops or yields. Input conditions and calibration data for

Hungr 1995, Haldenwang et al., 2006, and Iverson et al., 2010 experiments are provided in Table 1.

**Table 1. Input parameters for modelled experiments.**

| Variables | Hungr (1995) | Haldenwang et al. (2006) | Iverson et al. (2010) USGS Flume |
|---|---|---|---|
| Impoundment Height (m) | 30 | — | 1.9 |
| Volumetric Concentration (%) | 55 | 10 | 61.2 |





| Mixture Density (kg/m$^3$) | 2500 | 1165 | 2010 |
|---|---|---|---|
| $\tau_y$ (Pa) | 1500 | 21.3 | — |
| $d_s$ (mm) | 0.025 | 0.02-0.04 | 0.0625-5.0 |
| O'Brien Power Coefficient | 0.75 | 0.05 | |
| O'Brien Yield Coefficient | 6 | 9 | |
| O'Brien Viscosity Coefficient | 9.1 | 8 | |
| Hershel Bulkley Coef (k) (Pa.s$^n$) | — | 0.524 | |
| Hershel Bulkley Power (n) | — | 0.468 | |
| Yield Strength (Pa) | 1496.4 | 23.84 | |
| Dynamic Viscosity (Pa s) | 101.2 | 0.00316 | |
| | | | |
| Domain Length (m) | 2000 | 10 | 107.5 |
| Domain Width (m) | — | 0.015 | 2 |

295         \*Coefficients from O'Brien et al. (1993) yield strength and dynamic viscosity.

It is important to note that this paper only presents our general findings regarding USACE modelling capabilities. Subsequent publications will focus on the additional details for the provided flume experiments. The 2D HEC-RAS and AdH results for the Jeyapalan 1981 and Hungr (1995) dam breach simulations are included in Figure 2 (left and right respectively) representing conditions of dynamic unsteady conditions following an impoundment failure. The initial profile represents the 30 m high pre-dam

break conditions and the final profile is the analytical solution for the failure runout profile when the flow cesses. This analytical dam removal computes "run out length," which is the distance required for this high-energy mass flow to come to rest (i.e., the shear stress drops below the yield strength). Both numerical simulations compute run out lengths very similar to Hungr's (1995) result, and have similar final depth profiles. Both HEC-RAS and AdH sufficiently replicated the impoundment failure with to include unsteady runout, flow depth and velocity. These results were consistent with other publications based on this analytical

solution (e.g. Naef et al., 2005). Simulations were conducted using Bingham, Herschel-Bulkley, and O'Brien quadratic model with only Bingham rheology conditions presented with all cases exhibiting similar flow behaviour and results.

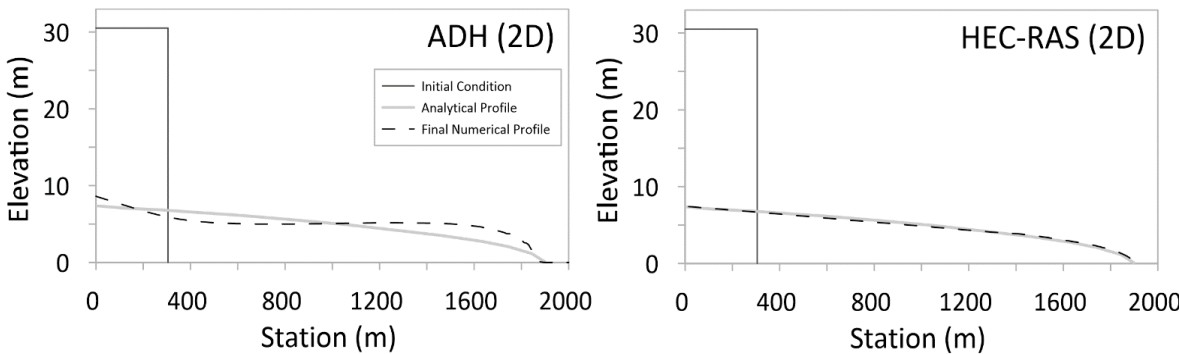

**Figure 2. Simulation of Hungr (1995) dam breach experiments with two-dimensional Adaptive Hydraulics.**



The results from the Haldenwang et al. (2006) simulations are included in Figure 3. HEC-RAS + DebrisLib using the 1D, finite-

difference, and 2D AdH finite element, unsteady flow (with a constant flow boundary condition) modelling frameworks to simulate

stages associated with multiple experimental flows with four different rheological assumptions. Results are plotted for sixteen

flows and depths from Haldenwang et al. (2006) 5-degree slope for the 10 m flume measured during a sequence of stepped, steady-

flow conditions for 10 percent kaolinite. Model results of the Haldenwang flume experiments are provided in Figure 3. In both

cases the non-Newtonian simulations compute fluid stage better than clear water flow. As shown overall good agreement between

flume experiments and model simulations, and reasonable agreement across all rheology closures presented to include Bingham,

HB. And O'Brien quadratic plotted with the Newtonian shallow-water results. Bingham and O'Brien provide the most accurate

representation of the flow and depths for the 5-degree flume conditions. This is consistent for both HEC-RAS and AdH simulation.

The differences between HEC-RAS and AdH results are likely due to a range of factors, to include how each model accounts for

initial conditions, model discretization, grid size, and time steps size. All rheology closure models simulated under predict depth-

flow curve at flow ranges smaller than approximately 7.5 L/s likely a result of several factors, to include the strain rate

approximation of $3u/h$ approximation, change in velocity profile, potential sediment deposition, or a combination. This will be

investigated more thoroughly in subsequent research. Interestingly all models did better at predicting transitional and turbulent

conditions versus laminar ((as shown with the red line in Figure 3). Only on previous study was found that used the Haldenwang

data set for comparison with a shallow-water model (Perez, 2017) but our results are consistent with those findings. Additional

model research is ongoing using this dataset aimed at betters quantifying turbulent behaviour at lower sediment concentrations

analogous to hyperconcentrated flows. This will also help better understand the deviations between model and flume results for

flow conditions less that about 5 L/s).





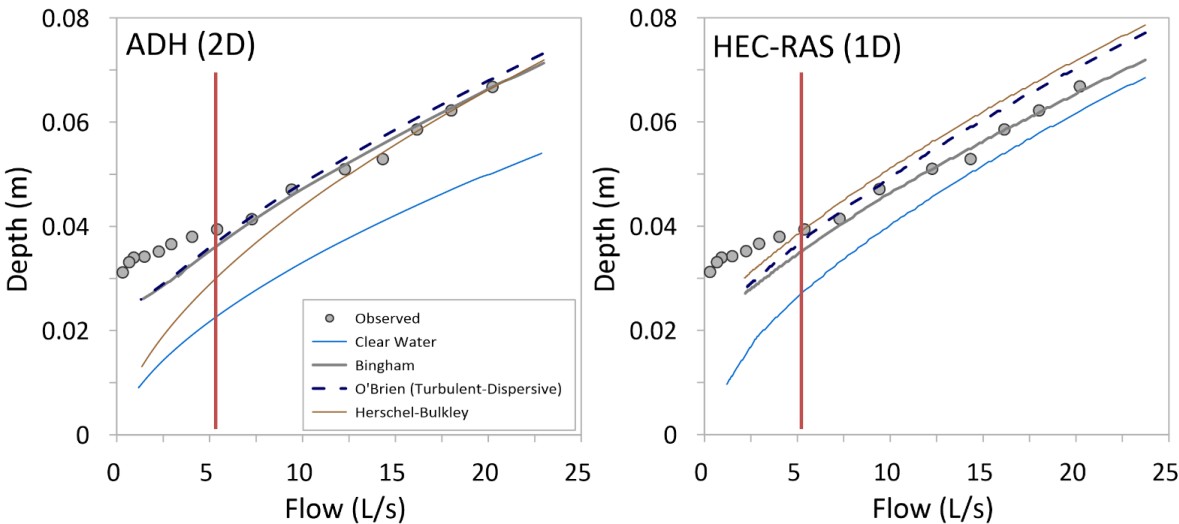

**Figure 3. Simulation of Haldenwang et al. (2010) flume experiment with one-dimensional HEC-RAS and two-dimensional AdH numerical model.**


AdH and HEC-RAS are validated with a representative experiment with a mixture of sand, gravel, and mud (silts and clays) and a section of bumpy concrete tiles. The comparison of AdH and HEC-RAS water levels compared to the measurements from one of the USGS experiments is presented in Figure 4. Both AdH and HEC-RAS water surface time series compared reasonably well

with pressure measurements. Differences in AdH and HEC-RAS results due to differences in the model formulation and numerical assumptions and solution methods. This demonstrates that the optimal rheological parameters will likely be different for different models. As shown in Figure 4 AdH more closely replicated the USGS flume results compared to the HEC-RAS results. Both models tend to do better at the upstream locations specifically at x = 2 m and x = 32 m. Model results continue to deviate from flume conditions slightly for x = 66 and x = 90 with more pronounced deviation seen in the HEC-RAS results. These differences

are the result of the flume slope being approximately 31 degrees which violates the shallow-water assumptions of most depth averaged models, including these. Surprisingly they still perform reasonable considering this limitation.

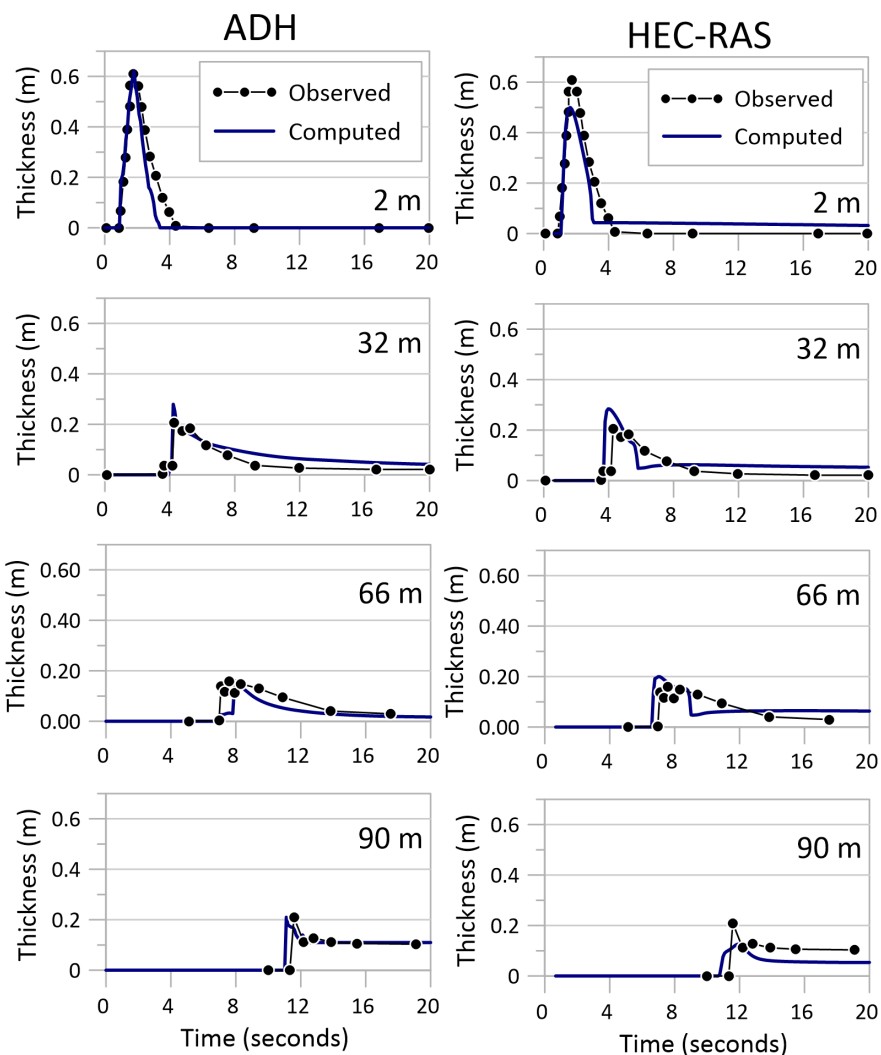

**Figure 4. Simulation of USGS flume experiments (Iverson et al., 2010) with two-dimensional HEC-RAS and two-dimensional AdH numerical models.**

## 5 Discussion

The post-wildfire non-Newtonian model library DebrisLib expands upon the quadratic model (O'Brien et al., 1993) to predict a range of post-wildfire sedimentation behaviour across multiple numerical model platforms. The flexible non-Newtonian library allows for prediction of a range of post-wildfire flood conditions demonstrated using a two shallow-water numerical models. We introduce a modification to existing non-Newtonian taxonomy and classification describing common non-Newtonian post-wildfire flows and demonstrated using both 1D and 2D shallow-water numerical models for a range of non-Newtonian flow conditions. To date, existing non-Newtonian models predict a limited range of flow conditions within a single modelling platform thus multiple models must be stitched together to address the full range of flow conditions. The process and approach diversity of the non-





Newtonian literature in particular make a modular computation library-based framework advantageous by consolidating the algorithms for each process. With a few common variables like depth, velocity, mixture density, and slope the model library can

be linked with most existing shallow-water Newtonian numerical models, regardless of dimension, library, or solution method, to predict non-Newtonian flows. Currently the model library is under development with planned released scheduled soon. Simulating a range of flow conditions across multiple non-Newtonian transport models will increase access to post-wildfire modelling capabilities and reduce uncertainty associated with comparisons from multiple independent model approaches. Furthermore, this research constitutes the first non-Newtonian simulations using USACE shallow-water models.

The flexibility will allow broader access to the non-Newtonian modelling capabilities for practitioners, managers, and researchers to improve understanding of the model limitations and operational modelling for post-wildfire emergency management and flood risk management. During development of the model library the team identified some key disadvantages and advantages associate with library-based frameworks in research and development. An overview of the advantages and challenges associated with development of a flexible library-based model framework are provided below in Table 3.

**Table 2. Advantages and challenges associated with development of Library-based frameworks.**

| Advantages | Disadvantages/Challenges |
|---|---|
| • Standardization and Transparency<br>• Leverages contributions<br>• Improve debugging and development<br>• Encourages scientific communication | • Requires continuous communication<br>• Research and development can take longer.<br>• Requires team commitment<br>• Focus on physical processes |

More specifically, the disadvantages and challenges identified are: 1) Modifications and changes during research and development require continuous communication between developers; 2) Because these changes and modifications require constant coordination research and development takes longer; 3) Requires commitment from all collaborators to ensure success; and 4) Requires

commitment from all collaborators to ensure success. Advantages include: 1) Standardization of computation approaches and closure between model frameworks; 2) Results in transparency in approach and assumptions; 3) Leverages contributions from multiple researchers and teams minimizing duplication; 4) Improves debugging and development production by expanding QA/QC capabilities; and 5) Encourages scientific communication between research teams by requiring explicit negotiations when addressing challenges and setbacks during development.

## 375  5 Conclusion

High-intensity wildfires remove vegetation, alter soils, modify subsurface root structures, purge organic soil, and create widespread hydrophobic soils. This increases runoff, erosion, and sediment transport which results in destructive flooding. Wildfire effects can generate gravity-driven surface runoff and erosion events that involve complex mixtures of water, ash, sediment, and entrained debris (i.e., destroyed upstream infrastructure, woody debris, and very large sediment clasts). Several other geophysical flows,

including lahars and mine tailing dam failures have similar non-Newtonian dynamics and require these closures to model them effectively. Hydrologic and hydraulic models that assess wildfire impacts on flood risk management need to develop and improve non-Newtonian numerical modelling capabilities. This work consolidates the state-of-the-practice in a modular non-Newtonian and sediment transport library. Then demonstrates that it can easily connect to multiple modelling platforms with different dimensionality, interfaces, and mesh requirements, and accurately predict commonly post-wildfire non-Newtonian flow

conditions. This modular library improves and enhances prediction capabilities to assist with planning, management, and mitigation in post-wildfire environments using practical science-based approaches and smart integrated numerical approaches.

### Acknowledgements

This research was supported by the research work unit of the US Army Corps of Engineer's Flood and Coastal Systems Storm Damage Reduction (F&CS) research program, specifically Flood Risk Management. Information for the F&CS program can be

found at https://chl.erdc.dren.mil/civil-works/mission-areas/flood-risk-management/.

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
