# Peer review of "A Modular, Non-Newtonian, Model, Library Framework (DebrisLib) for Post-Wildfire Flood Risk Management"

_Hydrology and Earth System Sciences, 2020_

## Referee Comment (RC1) · Anonymous Referee #1 · 10 Dec 2020

This manuscript introduces a computational library (DebrisLib) that can be used to help solve equations commonly used to represent the movement of geophysical flows. The library is designed to interact with models that utilize the shallow-water equations as a base, allowing for a range of flow resistance terms to be employed. The framework is tested by calling the library from two different models, HEC-RAS and AdH, and demonstrating that the models can reproduce results of experiments. The authors highlight several benefits of DebrisLib, including the ability for users to simulate geophysical flows that transition among a range of flow types from clear water flows to hyperconcentrated flows and debris flows. The work is motivated by an increasing need for models that are capable of simulating the range of flow conditions commonly

observed in post-fire flows. I agree that there is a need for improved modeling frameworks to simulate post-fire flows and see the benefit of a library like DebrisLib, which could add greater transparency, reproducibility, and flexibility to modeling efforts. However, I also have several comments and questions about methodology, assumptions regarding connections between flow rheology and flow classification, and how DebrisLib would be applied specifically to model post-fire flows, that I hope the authors find helpful in revising the manuscript.

General Comments:

1. The current capabilities of DebrisLib need to be more clearly articulated and separated from future improvements that are planned or still in development. There are hints that DebrisLib uses nondimensional thresholds to determine flow type and/or the relative importance of different flow resistance terms (lines 13-15, line 104). There are no nondimensional thresholds presented in this manuscript. In the model tests, it seemed like a particular rheology was assumed (i.e. line 305). This is an adequate test of the code and demonstrates that the library can be called from USACE shallow-water models, but falls short of testing a computational library that "...consolidates algorithms for each process and discriminates between these processes and algorithms with quantitative nondimensional thresholds." Similarly, there are places where it appears that sediment entrainment and/or deposition may somehow be involved but no erosion or deposition equations are presented.

2. The methods, especially specifics of how DebrisLib interacts with the parent code and how/why different rheological models were chosen for the test cases needs to be better described.

3. The flow classification system in Figure 1 needs to be more thoroughly described and justified. There are a number of assumptions regarding relationships between sediment concentration, grain size, and flow behavior. No definitions are given for mudflow, hyperconcentrated flow, or debris flow other than a general description in

line 68 that does not differentiate the three flow types. The differences between the four flow types defined in Figure 1 and mentioned throughout the text will need to be clarified. The relationship between the different rheological models included in the current version of DebrisLib and the flow classification scheme in Figure 1 needs to be described (lines 115-116 suggests there may be some relationship between the two but it is not clear).

4. DebrisLib makes it easier for modelers to employ different non-Newtonian rheological models but does not address the underlying issue that there is no objective way to define transitions in flow type. Moreover, it's not clear that transitioning from one rheological model to another as suggested by Figure 1 would be a practical way to simulate flows that transition from one flow type to another. Even when modeling the same flow through a watershed, it is not unreasonable to think that the optimal parameter values (e.g. for viscosity) will change as different rheological models are employed. There are no test cases that address issues related to flow transitions and no discussion of the limitations of the approach suggested by Figure 1. I could see multiple studies being dedicated to answering these types of questions so I don't expect them all to be answered here, but the limitations of the conceptual framework laid out in Figure 1 should be more thoroughly discussed, particularly within the context of how DebrisLib could be employed to simulate post-fire flows.

5. Post-fire hazards are mentioned as a motivating factor for developing DebrisLib and while I agree that DebrisLib could be quite useful for modeling post-fire flows, there is not a focus on fire throughout the manuscript and no test cases are related to post-fire flows. The title and motivation for the study should be revised or relevant details added about the effects of fire on hydrologic and geomorphic systems and how they lead to the generation of different types of geophysical flows.

Other Comments:

Line 11: Consider "non-Newtonian models" instead of "non-Newtonian algorithms"

Line 13: Consider "approaches to simulating non-Newtonian flows" rather than "approach to non-Newtonian simulations".

Line 13-15: But there are no nondimensional thresholds presented in this manuscript.

Line 23: I suggest ". . .post-wildfire flooding and debris flows" rather than only ". . .post-fire flooding."

Line 27-28: References needed for "Wildfires remove vegetation, reduce organic. . .." There is an implicit assumption here that the authors are referring to moderate or high severity fires so that should be mentioned.

Line 28: Fire can also reduce hydrophobicity and/or increase soil infiltration capacity in some cases.

Line 28: Consider "These changes" rather than "These processes"

Line 29: "Post-wildfire environments can cause. . ." Citations needed here.

Line 31: "In the years following a wildfire. . ." Citations needed here.

Line 33: This is very similar to what was already mentioned in line 27. I suggest removing this sentence and moving the references to the line beginning "Wildfire removes vegetation. . ."

Line 35: This sentence seems out of place here at the end of a wildfire-focused paragraph.

Line 38: It may be better to use a more general term such as "geophysical flows" rather than "non-Newtonian flows" in situations like this. The term "non-Newtonian" may imply the flow is being treated as an idealized fluid when some of the references cited treat the flow like a mixture where the stresses associated with the fluid and solid phases are evaluated separately (e.g. Iverson and Denlinger, 2001).

Line 45: Yes, but this could be motivated better with an expanded introduction. For example, past studies focused on post-fire flooding and debris flow initiation have demonstrated that debris flows are often triggered when rainfall intensity-duration (ID) thresholds are exceeded. Flows may transition from floods to debris flows (and everything in between) multiple times, even within the same rainstorm, depending on rainfall characteristics. Since rainfall ID thresholds for post-fire debris flow initiation may be on the order of a 1-year recurrence interval storm (e.g. Staley et al., 2020), transitions from floods or hyperconcentrated flows to debris flows are more likely to occur in recently burned areas relative to similar unburned areas.

Line 59: The manuscript briefly mentions some of the numerical algorithms but it would be helpful to describe the library and linkage architecture in more detail in the methods section.

Line 63: Impacts of wildfire extend beyond arid and semi-arid environments.

Line 64: Be more specific and include some reference(s) here. Recovery is defined in many different ways. Are you referring to hydrologic recovery, return of infiltration capacity or vegetation type/density to the pre-fire condition, etc?

Line 66: "wildfire-precipitation mix" is confusing. "wildfire-precipitation sequence"?

Line 70-73: This may fit better in the introduction to motivate the need for something like DebrisLib for post-fire flow modeling.

Line 72-73: Does this refer to the grain size distribution of the flow or of the hillslope and channel sediment? These are arguably factors that determine flow behavior given a particular runoff response, but there are many other factors that help determine how rainfall leads to runoff and sediment transport in a given burn area (burn severity, soil hydraulic properties, topography, ground cover).

Line 74: What processes?

Line 81: Add citation here for "dispersive model".

Line 92-97: Flow classifications need to be defined in some way. What is the difference between a hyperconcentrated flow and a mudflow? To what extent is the transition from one flow type to another dependent on sediment concentration versus grain size distribution? I appreciate the need for a conceptual model that helps motivate why libraries such as DebrisLib are useful, but I think this conceptual model overstates how much is known about these different types of flows and how they are related. It's also unclear if this is simply a conceptual model used to motivate the need for DerbisLib or if it is used in some way to determine flow rheology in this manuscript (i.e. for test cases presented later).

Line 100: Here and elsewhere, consider "post-wildfire flow responses" or "post-fire hydrologic responses" as a more general term since "flood" usually implies lower sediment concentration.

Line 104: "...determines flow conditions using non-dimensional parameters" Such as? What are these thresholds, how are they applied, and what flow types/conditions are they used to distinguish?

Line 105: "addresses hindered settling" suggests that entrainment and deposition are being modeled but this is not addressed elsewhere in the manuscript. Does settling still occur in flows classified as "debris flows"?

Line 106: It's not clear what is meant by "accounts for increased viscosities and mass density"

Line 109: This seems inconsistent with the prior line about hindered settling already being included.

Line 110: change to "requires computing"

Line 115: How is the flow classification made? How is flow classification related to rheology? This is a critical piece to the puzzle for simulating post-fire flows where flow classification may change rapidly in space and time, even during a single rainstorm.

Line 118-119: This is confusing because entrainment and deposition are not discussed here or elsewhere. If they are modeled, how are they modeled? If they are not modeled, then sediment concentration must be supplied as an input and the flow density, depth, etc, should already be consistent with that input sediment concentration so there would not be a need to modify flow properties.

Line 160: I get lost here. Is there a transition from a general discussion of the shallow water equations to the particular form of the equations that is used in HEC-RAS? I would remove the reference here to the diffusive wave approximation since it is not discussed in this section.

Line 181: Reference(s) needed. This is assuming that an effective fluid model is appropriate in the first place. In line 231, for example, "rheological models do not perform as well as the mixtures get more clastic".

Line 231: Reference(s) needed.

Line 234: change to "...yield strength with a geotechnical..."

Line 242: The numerical methods section would benefit from additional detail. It appears that this section mostly describes the shallow water parent code in general terms but does not describe specifics of how the mud and debris flow shear stress term, which is presumably computed by DebrisLib, is incorporated into the numerical solution of the parent code. Is this resistance term dealt with explicitly or implicitly and what specific methods are used? It is not uncommon for friction terms in shallow water models to promote instabilities or be stiff (e.g. Xia and Liang, 2018; Berthon et al., 2011) under certain circumstances so understanding how they were dealt with in the two models tested here could help readers determine if/how they may use DebrisLib in their models.

Line 263: How were simulations conducted? Were numerous rheological models tested? Was a single rheology tested (or some subset of the rheologies in Debris-

Lib) based on what was known about flow properties?

Line 264: change to "flume experiments"

Line 300: change to "ceases"

Line 301: Consider "This analytical solution can be used to compute the run-out length, which is...., and the final depth profile."

Line 303: Check wording here "with to include"

Line 305: "models" instead of "model"

Line 306: So the results in Figure 2 are using a Bingham rheology? Please clarify and include the model rheology in the caption for Figure 2.

Line 309: Missing a phrase to start the sentence "HEC-RAS+DebrisLib...."

Line 312: Some words are missing between "2006" and "5-degree"

Line 313: This sentence is essentially a repeat of the first sentence in the paragraph so it can be removed.

Line 316: The thought starting with "And O'Brien..." is a sentence fragment.

Line 325: "aimed at better"

Line 332: Somewhere in this sentence it would be good to remind the reader that you are referring to the USGS flume experiments or remove this sentence and start the paragraph with the next one.

Line 333-334: What rheology was assumed in these simulations? The number of simulations, parameter ranges involved in calibrations, and rheology used should be described in the methods section.

Line 333-334: "flow surface" would be more accurate than "water surface" and "water level" since the simulations presumably involve modeling a flow that contains both water

and sediment.

Line 346: I don't see what makes the library a "post-wildfire non-Newtonian library" as opposed to a "non-Newtonian library".

Line 349: The taxonomy shown in Figure 1 and described in the background section is not sufficiently detailed to allow a model user to objectively assign flow classifications based on the physical properties of the flow.

Line 356: Since the development of DebrisLib appears to be the primary contribution of this study, it seems important for the code to be released at the time of publication.

Line 376: Note the difference between fire intensity and fire severity. "Fire severity" is probably a better term here given that the focus is on the fire's impact on the vegetation and soil.

Line 376: Consider "...can create or enhance soil hydrophobicity" rather than "create widespread hydrophobic soils"

Line 383: What is the distinguishing factor here between "non-Newtonian" and "sediment transport" here. I didn't think there was a focus on sediment transport here.

Line 383: To justify "easily connect", it would be helpful to be more specific about how DebrisLib was connected to the parent code.

Line 385-386: This may be overstating the results since no post-fire flows were modeled. I do see how it would make it easier to use a single modeling framework to explore inundation scenarios associated with different types of geophysical flows that could be anticipated following a fire.

References

Berthon, C., Marche, F. and Turpault, R., 2011. An efficient scheme on wet/dry transitions for shallow water equations with friction. Computers & fluids, 48(1), pp.192-201.

[Figure]

Staley, D.M., Kean, J.W. and Rengers, F.K., 2020. The recurrence interval of post-fire debris-flow generating rainfall in the southwestern United States. Geomorphology, 370, p.107392.

Xia, X. and Liang, Q., 2018. A new efficient implicit scheme for discretising the stiff friction terms in the shallow water equations. Advances in Water Resources, 117, pp.87-97.

---

## Referee Comment (RC2) · Anonymous Referee #2 · 19 Jan 2021

This paper attempts to build a modular computation library by calling different modules to simulate the process of movement of post-fire debris flow. However, there is no convincing proof in this paper whether the models can describe every link of post-fire debris flow. Because the soil condition after a fire is different from the soil before a fire, which will result in lower infiltration rates, more rainfall splash erosion, and more severe gully erosion during movement of post-fire debris flow. Whether the model can truly reflect these scenarios requires more experimental data and field evidence. Comments can be found in below:

The post-fire debris flow is not only the debris flow with high concentration, but also dif-

[Figure]

ferent from the traditional debris flow due to the change of the physical and mechanical properties of the soil after fire. The consequence of this may affect the erosion of debris flow after fire and the amplification effect along the path, which is not well reflected in the model in this paper. And these changes need to be explained experimentally.

Reviewer agree with the author's consideration of the non-Newtonian property of debris flow in the shallow water equations. While the non-Newtonian property is closely related to the rheological property of post fire debris flow. Therefore, experiments and verification need to be added to make the model convincing.

The research object in this paper is post-fire debris flow. However, in order to verify the correctness of the model, the data of the three experiments of debris flow used in this paper are not from the experiment of post-fire debris flow. Even if the simulation results match the model, it cannot be proved that this model is suitable for the simulation of post-fire debris flow.

Post-fire debris flow often has more severe erosion ability than traditional debris flow. Equations (11-15) in this paper ignore the erosion ability to soil. This erosion ability is related to the physical and mechanical properties of soil after fire: Chen HX, Zhang LM. EDDA 1.0: integrated simulation of debris flow erosion, deposition and property changes. Geoentific Model Development, 2015, 8(3):829-844. Chang D S, Zhang LM, Xu Y, et al. Field testing of erodibility of two landslide dams triggered by the 12 May Wenchuan earthquake. Landslides, 2011, 8(3):321-332.

The discussion in this paper is not deep enough. The similarities and differences of erosion force and dynamic characteristics between post-fire debris flow and traditional debris flow should be discussed. Otherwise, the model will not be able to distinguish post-fire debris flow and traditional debris flow.

Table 2 does not address specific techniques, algorithms, efficiency advantages and disadvantages. It is suggested that the authors describe the advantages and disadvantages of the above content from a more objective perspective, which is the most

concerned by professionals.

Please further revise the reference format according to the requirements of the journal and add some latest literatures such as: Addison P, Oommen T. (2020). Post-fire debris flow modeling analyses: case study of the post-Thomas Fire event in California. Natural Hazards. 100(1). Cui, Y, Cheng DQ, Chan D. (2019). Investigation of Post-Fire Debris Flows in Montecito. ISPRS International Journal of Geo-Information. 8(1), 5. Staley DM, Negri JA, Kean JW, Laber JL, Tillery AC, Youberg AM (2017). Prediction of spatially explicit rainfall intensity–duration thresholds for post-fire debris-flow generation in the western United States. Geomorphology 278:149–162

Based on above comments, major revision is suggested.

---

## Referee Comment (RC3) · Anonymous Referee #3 · 22 Feb 2021

Review of "A Modular, Non-Newtonian, Model, Library Framework (DebrisLib) for Post-Wildfire Flood Risk Management" by Ian E. Floyd, Alejandro Sanchez, Stanford Gibson, and Gaurav Savant

This manuscript describes the theory and numerical modeling of DebrsiLib, a suite of algorithms intended to provide a framework for the modeling of non-Newtonian flows, such as mudflow, hyperconcentrated flow, and debris flow, with specific reference to post-fire debris flows. These algorithms are based upon the shallow water equations approach in HEC-RAS and AdH, with the "twist" being that of including modules that can handle the non-Newtonian properties of geophysical flows. To date, there is a

gap in our collective ability to model runoff-generated debris flows, hyperconcentrated flows, and mudflows, such as those that are common within and downstream of recently burned areas. This manuscript attempts to address this gap, and therefore represents an advancement of our scientific understanding and modeling of these hazardous events.

However, this manuscript is beset with issues that require significant revision. Primarily, the authors present the manuscript in the introduction and conclusions as work that can be used to address post-fire flood and debris-flow risk. However, the manuscript does not deal directly with the complexity of flows that emanate from recently burned areas, nor does it even begin to discuss how these algorithms can be used to characterize and reduce risk. Therefore, this manuscript overstates its applicability to post-fire flood risk. Therefore, substantial revision and repackaging of the manuscript is required. Suggestions for revision are described in the general and line-by-line-comments below. I recommend acceptance pending moderate to major revision.

General Comments:

1) The title of the manuscript indicates a model that can be used to address flood and debris flow risk management in and below recently burned watersheds. Outside of a few sentences in the introduction, discussion, and conclusions, this manuscript does not have any specific framework for application in recently burned areas (see comments below on initiation mechanisms and flow variability), nor does it discuss how it can be used in a risk management framework. As such, the title does not represent the overarching theme of the manuscript and should be changed to reflect the emphasis on modeling based on flume experiments and comparisons to Newtonian flow models (e.g., HEC-RAS and AdH).

2) The library includes modules that can deal with multiple flow types (hyperconcentrated, mud, grain, and debris flow, after Figure 1, and lines 67 – 69 and 93 – 95. However, these types of flows can often occur in sequence within and below recently

burned areas. For example, the initial surge is often debris flow (e.g. Kean et al., 2011), followed by periods of hyperconcentrated flow and/or subsequent debris flow surges. The code apparently treats each of these flow types independently, and therefore is not an accurate representation of real-world flow conditions which have significant effects on the manner in which the flows erode/deposit and interact with natural and engineered channel features, all of which influence the runout and inundation extents, velocities, and flow depths the determine a locations degree of risk.

3) It's unclear how/if these models deal with in-channel erosion and deposition, if at all. The authors refer to "settling" (lines 105) but I do not see anywhere else in the manuscript this is addressed.

4) Data used for calibration and testing are based upon flume experiments, where a known mass of sediment and known volume of water are released en masse. This design is intended to mimic flow initiation from landslide processes. Post-fire debris flows are most commonly initiated by runoff and severe erosion, and therefore the input of water and sediment would be more closely approximated using a hydrograph based approach where discharge and sediment concentrations vary rather significantly over a longer period of time (as compared to a landslide trigger). How does the difference in release mechanism in the flume(s) affect the model performance and accuracy when applied to a different initiation process?

5) I would like to see an expanded discussion of the modeling results. As a reader I am left wanting to hear more about how the model results compare to the calibration and testing data beyond what is presented in Figure 3. More maps, graphs, text explaining the results and relative accuracy and real-world implications of model results are warranted.

6) The manuscript touts DebrisLIB, but this suite of algorithms is apparently not available but "planned released scheduled soon" (line 355). As such, the manuscript should be streamlined to avoid overemphasis on the library of algorithms and focus solely the

development of the individual models and testing against flume data. Do not over-promise the results and applicability at this point in time. Once these models are re-viewed and published, it would be more appropriate to discuss real-world implications and applications in a separate "capstone-style" overview manuscript.

7) This is minor, but there are odd capitalizations throughout the manuscript that need to be fixed.

Line-by-line comments:

Title: Beyond the title and some lip service in the introduction and conclusions, this manuscript does not address risk management in and below recently burned areas. Therefore, the title and emphasis should be changed.

Manuscript L11: Algorithms are not non-Newtonian. Instead, "algorithms representing non-Newtonian flows." Also, "Non" should not be capitalized, here and throughout manuscript.

L28: should be "increase water runoff and erosion" or "increase runoff and sediment yield."

L63: recent fires in the southeastern US in 2016 and western Oregon in 2020 suggest the effects are not limited to arid and semi-arid environments.

L64: Decades to recover? Recover in coastal California, including Montecito, usually occurs in a period of a year or two. How are you defining "recovery?" Citations?

L92 – 97: It would be beneficial to the reader if the authors more formally defined the different types of flow beyond those in text and in Figure 1. Specifically, more discussion of the values represented in Figure 1 are warranted. Furthermore, it would be helpful if the authors discussed how a single flow "event" (e.g. January 9, 2018 in Montecito) may included multiple types of flows in sequence. Additionally, there is a weird mix of capitalizations in this section that should be fixed.

L118-119: Are you actually modeling erosion and deposition? Unclear to me. It's suggested that you modify sediment concentrations, but I do not see how entrainment and deposition are dealt with explicitly.

L160: "Shallow Water Equations" are capitalized here and not elsewhere. Be consistent.

L168 – 191: Equations are noted as #11 multiple times.

L254: "mildly nonlinear"?? It's either nonlinear or its not.

L287: should be "were used"

L296-297: Not really sure what is meant here by "regarding USACE modeling capabilities" and the flume experiments to be discussed in future papers.

L301 (and throughout): "Run out" is usually written as "runout" or "run-out." (I prefer the former over the latter)

L323: remove extra parenthesis

Figures: figure captions inadequately describe the figure. Any symbol/notation should be defined in figure caption.

Citations: There manuscript is lacking references to significant papers in post-fire debris flow and flooding research, further emphasizing the need to remove reference to post-fire debris flow and flood risk management from the title, introduction, and discussion/conclusions. No references for the D-Claw model developed by George and Iverson? The authors should at least present these papers and model by comparing and contrasting the differences between their approach and those of Iverson and George.

Here are the references:

George, D. L., & Iverson, R. M. (2014). A depth-averaged debris-flow model that includes the effects of evolving dilatancy. II. Numerical predictions and experimental tests. Proceedings of the Royal Society A: Mathematical, Physical and Engineering Sciences, 470(2170), 20130820. doi:10.1098/rspa.2013.0820

Iverson Richard, M., & George David, L. (2014). A depth-averaged debris-flow model that includes the effects of evolving dilatancy. I. Physical basis. Proceedings of the Royal Society A: Mathematical, Physical and Engineering Sciences, 470(2170), 20130819. doi:10.1098/rspa.2013.0819